# Clinical Efficacy of Biomimetic Bioactive Biomaterials for Dental Pulp Capping: A Systematic Review and Meta-Analysis

**DOI:** 10.3390/biomimetics7040211

**Published:** 2022-11-22

**Authors:** Liliana Argueta-Figueroa, Carlos Alberto Jurado, Rafael Torres-Rosas, Mario Alberto Bautista-Hernández, Abdulaziz Alhotan, Hamid Nurrohman

**Affiliations:** 1Tecnológico Nacional de México/Instituto Tecnológico de Toluca, Avenida Tecnológico s/n, Colonia Agrícola, Bellavista, La Virgen, Metepec 52149, Estado de México, Mexico; 2Consejo Nacional de Ciencia y Tecnología (CONACYT), Av. Insurgentes Sur 1582 Col. Credito Constructor, Alcaldia Benito Juarez 03940, Estado de México, Mexico; 3School of Dental Medicine, Texas Tech University Health Sciences Center, Woody L. Hunt School of Dental Medicine, 123 Rick Francis St, El Paso, TX 79905, USA; 4Centro de Estudios en Ciencias de la Salud y la Enfermedad, Facultad de Odontología, Universidad Autónoma “Benito Juárez” de Oaxaca, Av. Universidad s/n, Ex-Hacienda de Cinco Señores, Oaxaca 65120, Oaxaca, Mexico; 5Facultad de Medicina, Universidad Autónoma “Benito Juárez” de Oaxaca, Ex Hacienda de Aguilera s/n, Calz. San Felipe del Agua, Oaxaca de Juárez 68120, Oaxaca, Mexico; 6Dental Health Department, College of Applied Medical Sciences, King Saud University, Riyadh 11454, Saudi Arabia; 7Missouri School of Dentistry & Oral Health, A. T. Still University, Kirskville, MO 63501, USA

**Keywords:** resin-modified pulp capping biomaterials, TheraCal LC, dentin increment, clinical success, permanent teeth, primary teeth, pulp exposure

## Abstract

Recently, biomimetic bioactive biomaterials have been introduced to the market for dental pulp capping. This systematic review and meta-analysis aimed to determine any variation between the effect of using TheraCal LC and other bioactive biomaterials for pulp capping is different, as measured by dentin increment and clinical success. The risk of bias was assessed using the Risk of Bias 2 and Newcastle–Ottawa tools for randomized clinical trials and observational studies. A search for relevant articles was performed on five databases. Additionally, the quality of the included studies was assessed using the Grading of Recommendations Assessment, Development, and Evaluation (GRADE) criteria. A summary of individual studies and a meta-analysis were performed. The odds ratio of data from clinical success was combined using a random-effects meta-analysis. The meta-analysis results showed homogeneity between the studies (I^2^ = 0%). They revealed that the clinical success showed no differences between the patients who received TheraCal LC, light-cured calcium silicate-based biomimetic biomaterial, for dental pulp capping or the comparator biomaterials (*p* > 0.5). However, the certainty of the evidence was low to moderate due to the risk of bias in the included studies.

## 1. Introduction

Dental caries is a preventable disease; however, it is prevalent for carious lesions to reach deep areas of the dentin that are very close to the pulp chamber [1]. Minimally invasive dentistry involves selective caries removal, avoiding pulp exposure, and endodontic treatment [2]. When there is no pulpal exposure, but the proximity of pulpal tissue is evident, adequate pulp capping reduces pulpal irritation and induces the formation of repair dentin. However, direct pulp capping is indicated if pulp exposure of 1 mm in diameter or less occurs during carious tissue removal with adequate isolation. Traditionally, most clinicians considered there to be a low probability of success in direct pulp capping due to the contamination of the pulpal tissue [3]. Over time, pulp capping has a high clinical success rate in patients who meet the criteria to perform such treatment [4]. However, there has yet to be a consensus about the ideal biomaterial for pulp capping. A multi-national questionnaire-based survey performed by Stangvaltaite et al. [5] found that dentists who read scientific articles on the subject would consider, in the first instance, performing direct pulp capping instead of endodontic treatment (Odds Ratio = 2.1, 95% confidence interval, 95% CI = 1.3, 3.2). This study also revealed that about 50% of dentists use self-cured or light-cured calcium hydroxide for direct pulp capping. Ali et al. [6] found that the biomaterial preferred was calcium silicate-based for direct and indirect pulp capping procedures.

Synthetic biomimetic, bioinspired, and bioactive biomaterials seek to solve problems or replace structures by copying what occurs in nature [7]. A recent biomimetic biomaterial has been introduced to the market, a light-cured resin modified with tricalcium silicate filler particles, commercially available as TheraCal LC (Bisco, Schaumburg, IL, USA) [8]. The TheraCal LC biomaterial mainly consists of Portland cement type III, Poly(ethylene glycol) dimethacrylate, Bis-GMA, and barium zirconate [9]. An in vitro study comparing different biomaterials as pulp caps reported that TheraCal LC showed adequate shear bond strength (SBS) and the internal marginal adaptation of the material to dentin; these properties were similar to those exhibited by alternative none light-cured calcium silicate biomaterials [10]. On the other hand, an in vitro study determined that TheraCal LC’s capability to alkalize the pH 11 to 12 during the first 72 h decreased until two weeks, after which the pH remained slightly alkaline. Furthermore, TheraCal LC showed low porosity and solubility in comparison with other calcium hydroxide and calcium silicate-based biomaterials; due to that, it has been proposed that the ability to release a moderate but constant amount of calcium ions could promote the nucleation of calcium phosphate deposits, which lead to bridge dentin formation [11]. However, there are concerns about the toxicity of TheraCal LC in the pulp tissue due to their organic components and if regarding complete polymerization occurs [12]. Resin-containing biomaterials such as TheraCal LC may shift the pulp response towards the inflammatory reaction while altering the regeneration process [9].

Therefore, a systematic review and meta-analysis were undertaken to compare TheraCal LC’s effectiveness and other bioactive pulp capping biomaterials regarding dentin increment and clinical success. 

## 2. Materials and Methods

This systematic review was conducted according to the PRISMA (Preferred Reporting Items for Systematic Reviews and Meta-Analyses) guidelines and the Cochrane handbook [13,14].

### 2.1. Protocol Registry

The protocol for this systematic review was registered on the International Prospective Register of Systematic Reviews (PROSPERO) database (CRD42022361656). 

### 2.2. Eligibility Criteria and Participant Characteristics of Studies

The eligibility criteria were defined considering the PICO (Population, Intervention, Comparison, and Outcome) strategy. The studies included in the systematic review were clinical trials and observational studies that answered the research question, and the full text of the articles must be in English or Spanish. The search was performed without restriction of publication time of the studies. The studies excluded were case reports, case series, letters, comments, short communications, animal studies, in vitro studies, and literature reviews. 

### 2.3. Search Strategy and Databases Used

The research strategy and the databased used for the search are shown in Table 1. The search was carried out in September 2022.

### 2.4. Risk of Bias in Individual Studies and Quality Assessment

Two researchers (RTR and LAF) performed the risk of bias assessment of the included studies. The disagreements were resolved by consensus of the research group. The Newcastle-Ottawa quality assessment scale (NOS) [15] or the Risk of Bias 2 (RoB 2) [16] were used for observational studies (OS) and randomized clinical trials (RCTs), respectively. The figure of the risk of bias assessment was built as in previous research using the RoB 2.0 Excel implementation [17]. Additionally, the quality of included studies was assessed using the Grading of Recommendations Assessment, Development, and Evaluation (GRADE) criteria [18]. 

### 2.5. Study Selection

As in previous systematic reviews, the eligibility of the included studies was determined by reading the title and abstract of each record identified by the search. Subsequently, the full texts of the selected articles that could be reviewed in-depth were acquired [19]. Finally, when the reviewed studies did not fully meet the eligibility criteria, these were excluded with reasons.

### 2.6. Data Collection Process and Data Items

A standardized worksheet was prepared for the registration of the relevant data from the included studies, such as study design, population (n), main (clinical success and dentin thickness), and secondary outcomes. The corresponding authors of the included studies were contacted via email for missing data or additional details.

### 2.7. Data Synthesis

A table was built to summarize quantitative findings for each main outcome from the included studies. The heterogeneity between the measured effects from the studies was evaluated. The data were grouped according to primary or permanent dentition, direct or indirect pulp capping, and follow-up time. The heterogeneity in the methodologies of the included studies was thoroughly analyzed; in order to perform the quantitative synthesis. 

### 2.8. Meta-Analyses

For the meta-analyses, the odds ratio was calculated from the individual studies, and these were combined using a random-effects meta-analysis with Review Manager (RevMan Version 5.4, The Cochrane Collaboration, 2020). On the other hand, the Test for Funnel Plot Asymmetry and the Likelihood-ratio Test for publication bias were performed with the results of clinical success from all the included studies with esc, metafor, and weightr packages of the R software (R Development Core Team, 2011. Version 4.2.0).

## 3. Results

### 3.1. Selection and Characteristics of the Studies

The initial search yielded a total of 520 records from the databases. Subsequently, after removing duplicates, 284 records remained. Then, 19 full-text articles were retrieved for eligibility. Of these, 5 studies were excluded with reasons (Table 2). Thus, 14 studies were included in the present review: 2 observational studies (OS) and 12 randomized clinical trials (RCT). The study selection process is detailed in the PRISMA flow diagram (Figure 1).

### 3.2. Results of Individual Studies

In total, 2 OS [20,21] and 12 RCTs [22,23,24,25,26,27,28,29,30,31,32,33] were found. Of these, four studies evaluated light-cured calcium silicate-based biomimetic biomaterial (TheraCal LC) in primary teeth [22,23,24,25] (Table 3) and ten studies in permanent teeth [20,21,26,27,28,29,30,31,32,33] (Table 4). No non-RTC were found. In contrast, different outcomes reflecting the clinical efficacy of TheraCal LC treatment were found in the included studies as clinical success and dentinal bridge formation. The studies of interest considered a clinical success as the absence of pain, fistula, tenderness, and abnormal mobility or radiographical signs of failure, such as radiolucency in the periapical zone. Dentin increment was measured in digitalized radiographs of permanent teeth using CorelDRAW X3 software [22] or ImageJ software [29].

Moreover, secondary outcomes such as pain, microbiological performance, and histological findings were also found across the studies. Regarding pain after the treatment of primary teeth, two studies evaluated it. Erfanparast et al. [23] reported that only 1/37 patients had pain in the MTA group with direct pulp capping, while Abdelhafez et al. [25] reported that 2/10 patients experienced pain after the partial pulpotomy with TheraCal LC. In contrast, none had pain in the MTA group. Furthermore, two studies assessed the pain after direct pulp capping in permanent teeth. Cengiz et al. [26] reported that 2/15 patients had pain in the Dycal group and 3/15 patients in the TheraCal LC group. Sameia et al. [29] reported that there was no difference in pain between the patients treated with TheraCal LC and MTA. Two studies determined the pain for indirect pulp capping in permanent teeth; Rahman et al. found no pain in the patients of the TheraCal LC group and Mahapatra et al. [33]. These results suggest that the pain after the direct pulp capping is more likely with TheraCal LC treatment, whereas there was no difference for indirect pulp capping.

Due to the clinical success assessment depends on the appreciation of signs and symptoms, evaluation of inter-examiner reliability is crucial. The present review found that some studies assessed inter-examiner reliability using the kappa test. Those studies were Erfanparast et al. [23] (one examiner; inter-examiner test: k = 0.93), Rahman et al. [32] (2 examiners inter-examiner test: k = 0.94), Sahin et al. [24] (2 examiners inter-examiner test: k = 0.88, intra-examiner test: k = 0.94), and Abdelhafez et al. [24] (2 examiners inter-examiner test: k = 0.756, intra-examiner test: k = 0.988). Eleven included studies were not performed in the reliability test. 

Only one of the included studies measured dentin bridge formation. Menon et al. [22] found that the dentin increment using MTA and TheraCal LC as indirect capping from the baseline until six months was not statistically significant. Thirteen studies evaluated the clinical success of pulp capping. Of these, two studies were performed direct pulp capping in primary teeth, two studies were performed indirect pulp capping in primary teeth, four studies were performed direct pulp capping in permanent teeth, four studies were performed indirect pulp capping in permanent teeth, one study performed direct and indirect pulp capping in permanent teeth reporting its results for separate, and one study performed direct and indirect pulp capping in permanent teeth reporting combined results. 

Of the included studies, only one article evaluated the pulp capping performance in a pulp exposure model in healthy teeth. Bakhtiar et al. [27] performed an RCT of 8 weeks of follow-up. Clinical sound third molars were submitted to a partial pulpotomy and direct capping. At the end of the follow-up, all teeth from all groups exhibited clinical success; however, two patients in the TheraCal LC group exhibited pain, while the patients treated with Biodentine and ProRoot MTA did not experience any pain. Regarding the histologic findings, none of the teeth treated with Biodentine showed disorganization of the entire pulp tissue, whereas 22.2% of TheraCal LC-treated and 44.4% of ProRoot MTA–treated cases showed disorganization of the entire pulp tissue. All teeth in the Biodentine group showed a complete dentin bridge formation; in contrast, the teeth in the TheraCal LC and ProRoot MTA groups showed complete dentin bridge formation only in 11% and 56% of the teeth, respectively. The other studies examined in the systematic review were performed in teeth with deep caries lesions. The findings revealed that the latter design better-emulated patients’ clinical conditions requiring direct or indirect pulp capping.

For the other side, the most common comparators were self-cured calcium hydroxide (Dycal and Prevest Cal LC, Life Kerr AC), which were used in eight included studies. Other comparators used were light-cured calcium hydroxide based-biomaterials (Calcihyd, Calcimol LC), light-cured calcium hydroxy phosphate (Biner LC), calcium silicate based-biomaterials (ProRoot MTA, BioMTA+, and Biodentine).

### 3.3. Risk of Bias and Quality Assessment

At the overall bias of the included studies, 14.3% exhibited low risk, 57.1% had some concerns, and 28.6% had high risk; the main problems in the studies were reporting (85.7%), randomization process (57.1%) and deviations from intended interventions (42.9%). The risk of bias for individual studies is shown in Figure 2. Moreover, the certainty of the evidence was low to moderate due to the risk of bias in the included studies, as shown in Table 5.

### 3.4. Meta-Analysis

The data collected from the selected studies were grouped according to primary or permanent dentition, direct or indirect pulp capping, and follow-up time; all the meta-analyses performed showed an absence of heterogeneity (I^2^ = 0%). The meta-analysis of the clinical success of the light-cured calcium silicate-based biomimetic material for direct dental pulp capping at six months suggested that the clinical success rate was no different between the teeth treated with light-cured calcium silicate or those treated with other calcium-based materials (Odds Ratio 95% CI = 0.86 [0.46, 1.60]; *p* = 0.91), see Figure 3A.

The meta-analysis of the clinical success of the light-cured calcium silicate-based biomimetic material for indirect dental pulp capping at six months suggested that the clinical success showed no differences between the teeth treated with light-cured calcium silicate or other calcium-based materials (Odds Ratio 95% CI = 3.19 [0.32, 32.32]; *p* = 0.33), as shown in Figure 3B. This tendency was observed at 12 months (Odds Ratio 95% CI = 1.97 [0.35, 10.94]; *p* = 0.44), as shown in Figure 3C.

Meta-analysis of included studies with primary teeth could not be performed due to methodological heterogeneity.

The result of the Test for Funnel Plot Asymmetry was z = −0.1745, *p* = 0.8614, limit estimate (as estimated coefficient’s standard error, SEi ≥ 0): b = 0.0965 (95% CI = −0.2061, 0.1724), as shown in Figure 4. Furthermore, a likelihood-ratio test was conducted comparing the adjusted model including selection to its unadjusted random-effects, counterpart. The Vevea and Hedges Weight-Function Model results in a likelihood-ratio of χ2 = 0.6282, *p* = 0.4279. All this suggests that there was no publication bias.

## 4. Discussion

The findings of the current meta-analyses of the effect of TheraCal LC on direct and indirect pulp capping in permanent teeth revealed no difference between this bioactive biomaterial and the controls. We could not perform a meta-analysis of primary teeth due to the methodological heterogeneity of the studies and the different follow-up time intervals. However, the tendency of the results of the individual studies is similar to those observed with permanent teeth.

The evidence indicates that TheraCal LC causes more pain than self-cured calcium hydroxide, MTA, or Biodentine, especially in patients who underwent direct pulp capping. Nevertheless, it is likely that the pain eventually diminishes, and the pulp tissue remains vital. Consequently, TheraCal LC does not seem to decrease clinical success, at least during the follow-up period of the included studies [3].

The MTA used as a comparator in the included studies was mainly from two commercial brands, MTA Angelus and ProRoot MTA. MTA Angelus has a higher amount of small bismuth oxide particles compared to ProRoot MTA. The small particles of this type of cement can favor the sealing of the dentinal tubules and improve the sealing ability of the biomaterial [34]. However, one of the significant disadvantages of MTA is its long setting time. In contrast, Biodentine has a smaller particle size than MTA and a short setting time [35]; however, no relevant improvement in clinical success has been reported using Biodentine for dental pulp capping [36].

Regarding sealing ability, an in vitro study showed that TheraCal LC had a significantly higher internal gap at the dentinal surface when compared to MTA and Biodentine; however, MTA and Biodentine had comparable results in sealing ability [37]. Concerning the microbiological effect, a randomized clinical study was conducted using permanent carious molars treated with indirect capping and found that the reduction of the bacterial count in both *Streptococcus mutans* and *Lactobacillus* was using Biner LC (light-cured hydroxy calcium phosphate) or TheraCal LC [38]. Moreover, in vivo subcutaneous implantation test performed in rats showed irregular collagen formation and tissue inflammation; such study informed that inflammation in the liver and kidney tissue of the rats was more severe with the use of TheraCal LC, possibly attributable to the light-cured compounds [39].

It has been reported that only age (under 40 years old), exposure site, and capping biomaterial had significant effects on the pulpal survival rate. Studies suggested that the clinical success of pulp capping depends mainly on using a proper technique of disinfection, asepsis, and rubber dam isolation. Furthermore, an adequate follow-up of the patients, with periodic clinical appointments to check the pulp vitality and the integrity of the restorations, can be crucial for the success of this treatment in the long term [40].

One of the disadvantages of TheraCal LC is that a thin enough layer must be applied so that the light can lamp-polymerize to the deepest part of the biomaterial. The thickness of the specimens should be 0.65 mm [12]. TheraCal LC PT is a new biomaterial that recently came onto the market that claims to solve this problem through dual curing. However, no clinical studies have been published so far, so the efficacy of these two biomaterials cannot yet be compared. 

Nanotechnology has been used as a novel approach to improve bioactive dental materials [41,42]. Micro-nano materials with regenerative dentin properties have also been developed for dental pulp capping [43]. Moreover, some studies suggested that nano-hydroxyapatite promotes dense reparative dentin formation, produces complete dentinal bridges, and increases cellular and vascular response [44,45]. However, to our knowledge, calcium-based biomaterials and no light-cured calcium silicate-based (MTA and Biodentine) remain the best options for dental pulp capping.

For the last two decades, the publications of basic research, observational studies, and clinical trials have increased, and their quality has been improved with the reporting guidelines [46,47]. RTCs are considered the best evidence to unravel medical inquiries. However, some basic research results are not translated into clinical practice [48]. Nevertheless, several research studies have shown opposing results even though their characteristics are similar or use different outcomes to evaluate the effect direction of the intervention. Consequently, the efforts for evidence-based medicine aimed to assess the validity of studies through a systematic review [49].

## 5. Conclusions

The current evidence of TheraCal LC for direct and indirect pulp capping in permanent teeth showed no difference between this bioactive biomaterial and the controls. The clinical efficacy of TheraCal LC for primary teeth is unclear. Moreover, the certainty of the evidence was low to moderate due to the risk of bias in the included studies.

## Figures and Tables

**Figure 1 biomimetics-07-00211-f001:**
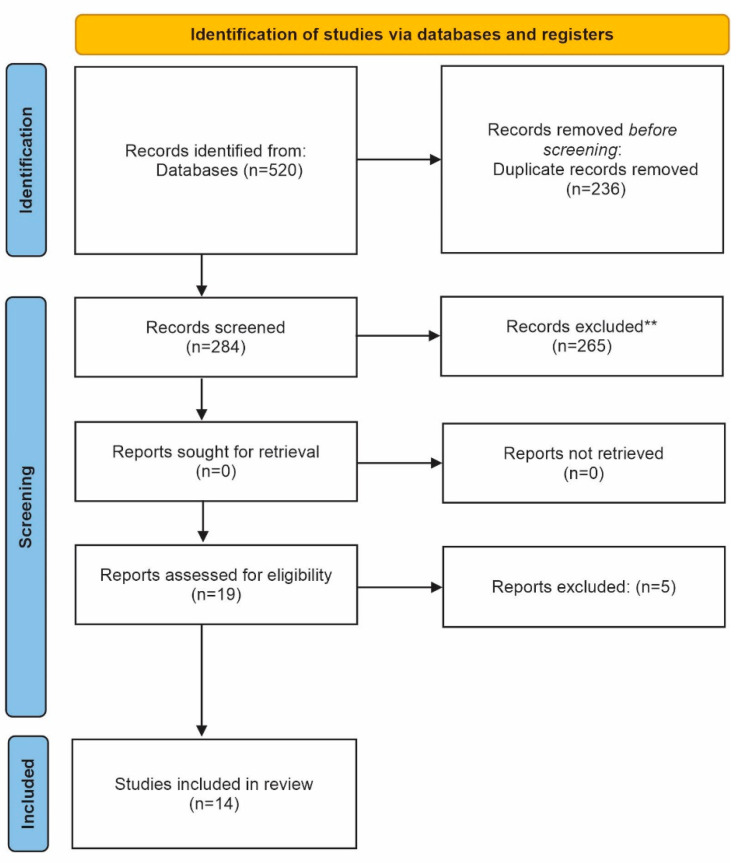
Prisma flow diagram of the selection process of the studies included in the systematic review.

**Figure 2 biomimetics-07-00211-f002:**
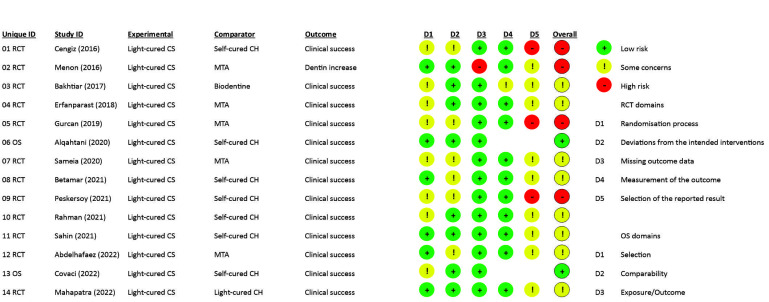
Risk of bias of the studies included in the systematic review.

**Figure 3 biomimetics-07-00211-f003:**
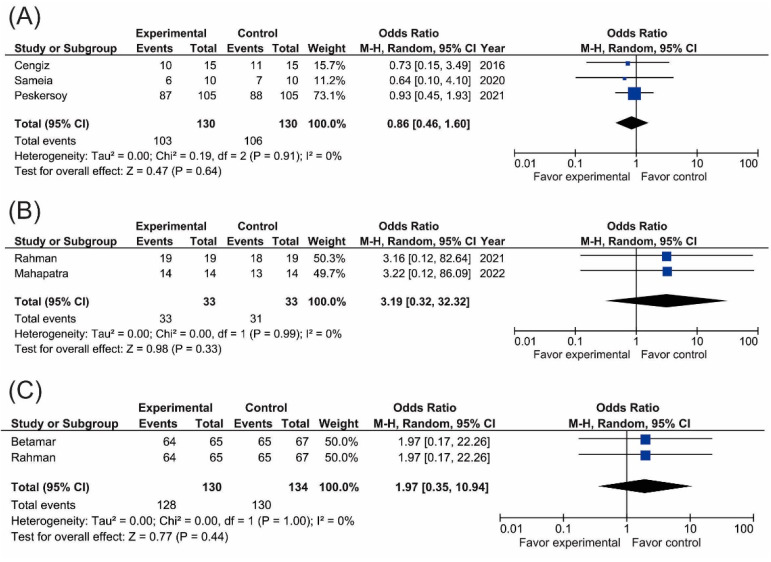
Meta-analysis. Forest plots of clinical success of light-cured calcium silicate based-material (TheraCal LC); (**A**) Direct pulp capping in permanent teeth at 6 months, (**B**) Indirect pulp capping in permanent teeth at 6 months, and (**C**) Indirect pulp capping in permanent teeth at 12 months.

**Figure 4 biomimetics-07-00211-f004:**
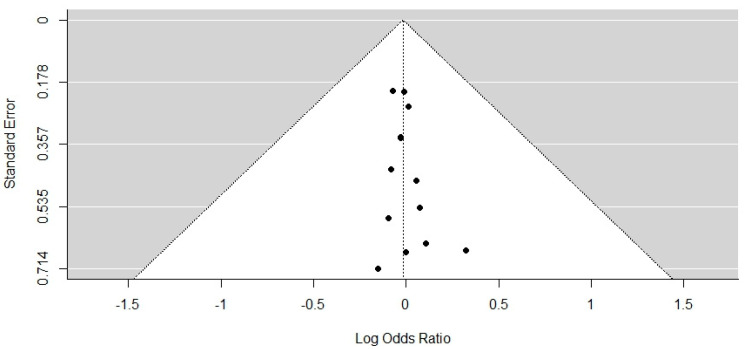
Funnel plot of the included studies.

**Table 1 biomimetics-07-00211-t001:** PICO and search strategy for the systematic review of the light-cured calcium silicate-based biomaterial.

Population	Patients under direct or indirect pulp capping therapy in primary or permanent teeth
Intervention	TheraCal LC
Comparator	Other bioactive pulp capping biomaterials
Outcomes	Dentin increment and clinical success (the absent of pain, fistula, tenderness, and abnormal mobility or radiographical signs of failure such as radiolucency in the periapical zone)
Study design	Randomized clinical trials (RTCs), non-randomized clinical trials (non-RTCs) and observational studies (OS)
Languages	English and Spanish
Electronic databases	PubMed, Google Scholar, Scopus, Web of Science, and Clinical Trials
Focused question	Which biomimetic bioactive biomaterials is more effective for dental pulp capping in terms of dentin increment and clinical success?
Registers found for each database	Algorithms used for search strategy adapted for each database
Scholar Google *n* = 254	(“Resin-based Tricalcium Silicate” OR “light-cured calcium silicate-based” OR “resin-modified Portland cement-based material” OR “TheraCal”) +”clinical study”
PubMed *n* = 87	((Resin-based[All Fields] AND (“calcium silicate”[Supplementary Concept] OR “calcium silicate”[All Fields])) OR (light-cured[All Fields] AND (“calcium”[MeSH Terms] OR “calcium”[All Fields]) AND silicate-based[All Fields]) OR (resin-modified[All Fields] AND Portland[All Fields] AND cement-based[All Fields] AND material[All Fields]) OR (“TheraCal”[Supplementary Concept] OR “TheraCal”[All Fields] OR “theracal”[All Fields])) AND (“dental pulp capping”[MeSH Terms] OR (“dental”[All Fields] AND “pulp”[All Fields] AND “capping”[All Fields]) OR “dental pulp capping”[All Fields] OR (“pulp”[All Fields] AND “capping”[All Fields]) OR “pulp capping”[All Fields])
Clinical Trials *n* = 4	Calcium silicate|Completed Studies|Pulp capping|clinical success Applied Filters: Completed
Scopus *n* = 93	TITLE-ABS-KEY ((“Resin-based Tricalcium Silicate” OR “light-cured calcium silicate-based” OR “resin-modified Portland cement-based material” OR “TheraCal”) + “pulp capping”)
Web of Science *n* = 82	(ALL = ((“Resin-based Tricalcium Silicate” OR “light-cured calcium silicate-based” OR “resin-modified Portland cement-based material” OR “TheraCal”))) AND ALL = (“pulp capping”)

**Table 2 biomimetics-07-00211-t002:** Studies excluded with reasons.

Reason of Exclusion	Reference
Chinese language	Ma XH, Xie NN. Application of different biomaterials in the preservation of vital pulp in carious deciduous teeth: A prospective, single-center, randomized, controlled clinical trial. *Chinese Journal of Tissue Engineering Research.* 2017;21(22):3494–3500.
Ex-vivo study	Kamal EM, Nabih SM, Obeid RF, Abdelhameed MA. The reparative capacity of different bioactive dental materials for direct pulp capping. *Dent. Med. Probl.* 2018 Apr-Jun;55(2):147–152. doi: 10.17219/dmp/90257. PMID: 30152617.
Full-text does not available	Parvin MK, Moral AA, Shikder ZH, Alam MS, Bashar AM. Evaluation of Radiological Outcomes of Theracal Light Cured (TLC) And Calcium Hydroxide As Indirect Pulp Capping Agents In The Treatment Of Deep Carious Lesion Of Permanent Molar Teeth. *Mymensingh Med. J.* 2018 Oct;27(4):859–865. PMID: 30487505.
	Mahapatra J, Nikhade PP, Belsare A. Comparative evaluation of the efficacy of theracal lc, mineral trioxide aggregate and biodentine as direct pulp capping materials in patients with pulpal exposure in posterior teeth-an interventional study. *International Journal of Pharmaceutical Research.* 2019;11(2):1819–1824.
Inadequate comparator	Yazdanfar I, Barekatain M, Zare Jahromi M. Combination effects of diode laser and resin-modified tricalcium silicate on direct pulp capping treatment of caries exposures in permanent teeth: a randomized clinical trial. *Lasers Med. Sci.*. 2020 Oct;35(8):1849–1855. doi: 10.1007/s10103-020-03052-9. Epub 2020 Jun 11. PMID: 32529588.

**Table 3 biomimetics-07-00211-t003:** Characteristics of the individual studies and their results.

Primary Teeth					
Reference	Participants	Type of Pulp Capping	Groups	Main Outcome	Secondary Outcomes
**Menon (2016)** **Randomized clinical trial**	Participants *n* = 21 Age: 4–7 yrs-old Primary molars with deep caries *n* = 43	Indirect	A. TheraCal B. MTA	**Dentin increment** At 6 months (mean ± SD mm) A = 0.154 ± 0.011; B = 0.151 ± 0.021	ND
**Erfanparast (2018)** **Split mouth randomized clinical trial**	Participants *n* = 46 Age: 5–7 yrs-old Primary molars with deep caries *n* = 92 Participants at the end of follow-up *n* = 37 Teeth at the end of the follow-up *n* = 74 Dropouts *n* = 9 patients	Direct	A. TheraCal B. ProRoot MTA	**Clinical success** At 12 months: A = 34/37; B = 35/37	Pain
**Sahin (2021)** **Randomized clinical trial**	Participants *n* = 109 Age: 5–9 yrs-old Primary molars with deep caries *n* = 109 Dropouts to the end of the follow-up: *n* = 8 Physiological exfoliation: *n* = 10 Participants at the end of the follow-up *n* = 91	Indirect	A. TheraCal B: Biodentine C. Dycal	**Clinical success** At 6 months: A = 35/36; B = 36/36; C = 37/37 At 12 months: A = 34/35; B = 35/35; C = 37/37 At 18 months: A = 28/28; B = 34/34; C = 33/33 At 24 months: A = 28/28; B = 30/30; C = 30/30	Histological evaluation
**Abdelhafez (2022)** **Split mouth randomized clinical trial**	Participants *n* = 30 Age:3–6 yrs-old Primary molars with deep caries *n* = 60	Direct: A Partial pulpotomy B Complete pulpotomy	A1. TheraCal A3. MTA B1. TheraCal B3. MTA	**Clinical success** At 6 months: A1 = 10/10; A2 = 10/10; B1 = 10/10; B2 = 10/10 At 12 months: A1 = 10/10; A2 = 10/10; B1 = 10/10; B2 = 10/10 At 15 months: A1 = 8/10; A2 = 8/10; B1 = 9/10; B2 = 9/10	Pain

ND: No data available. TheraCal (light-cured calcium silicate) Bisco Inc., Schaumburg, IL, USA. Dycal (self-cured calcium hydroxide) Dentsply-Sirona, Charlotte, NC, USA. Biner (light-cured hydroxy calcium phosphate). META-BIOMED, Chungcheongbuk-do, Korea. Calcihyd (light-cured calcium hydroxide) Dr. Roberts’, Istanbul, Turkey. Prevest Cal (light-cured calcium hydroxide) DenPro, Brussels, Belgium. MTA-Angelus, (mineral trioxide aggregate) Londrina, Paran, Brazil. Biodentine (self-cured calcium silicate) Septodont, Saint-Maur-des-Fosse’s France. ProRoot aggregate) Dentsply Tulsa, Johnson City, TN, USA. MTA PPH MTA (mineral trioxide aggregate) PPH CERKAMED, Stalowawola, Poland. BioMTA+ (modified tricalcium silicate) Cerkamed, Stalowa Wola, Poland. Calcimol LC (light-cured calcium hydroxide) Voco GmbH, Cuxhaven, Germany. Life Kerr AC (self-cured calcium hydroxide) Kerr, Orange, CA, USA.

**Table 4 biomimetics-07-00211-t004:** Characteristics of the individual studies and their results.

Permanent Teeth					
**Cengiz (2016) Randomized clinical trial**	Participants *n* = 60 Age:18–41 yrs-old Premolars and molars with deep caries (*n* = 60)	Direct	A. TheraCal B. Dycal	**Clinical success** At 6 months: A = 10/15; B = 11/15	Pain
**Bakhtiar (2017)** **Randomized clinical trial**	Participants *n* = 27 Age:18–32 yrs-old Sound human maxillary and mandibular third molars scheduled for extraction *n* = 27	Direct	**Clinical success** A. TheraCal B. Biodentine C. ProRoot MTA	**Clinical success** At 8 weeks: A = 9/9; B = 9/9; C = 9/9	Pain Histologic Findings
**Gurcan (2020)** **Randomized clinical trial**	Participants *n* = 95 Age: 4–15-yrs-old Second primary molars with deep caries *n* = 135 First permanent molars *n* = 160	Indirect	A. TheraCal B: ProRoot MTA C. Dycal	**Clinical success** At 24 months: A = 87.8%; B = 94.4%; C = 84.6%	ND
**Alqahtani (2020)** **Observational study**	Participants *n* = 120 Age: 23–75 yrs-old Clinical records from: 2012–2015. Teeth with deep caries *n* = 148	Direct and Indirect	A. TheraCal B. Dycal	**Clinical success** Direct pulp capping at 3 months: A = 9/13; B = 6/12 Indirect pulp capping at 3 months: A = 58/66; B = 53/57	ND
**Sameia (2020)** **Randomized clinical trial**	Participants *n* = 20 Age: 17–35 yrs-old Teeth with deep caries *n* = 20	Direct	A. TheraCal B. MTA PPH	**Clinical success** At 1 week: A = 10/10; B = 10/10 At 1 month: A = 10/10; B = 10/10 At 3 months: A = 10/10; B = 10/10 At 6 months: A = 6/10; B = 7/10 **Dentin increment** At 6 months (mean ± SD mm): A = 1.765 ± 0.436; B = 1.856 ± 0.420	Pain
**Betamar (2021)** **Randomized clinical trial**	Participants *n* = 130 Age: 8–55 yrs-old Premolars and molars teeth with deep caries *n* = 200	Indirect	A. TheraCal B. Dycal C. Biner	**Clinical success** At 12 months: A = 64/65; B = 65/67; C = 66/68	ND
**Peskersoy (2021) Randomized clinical trial**	Participants *n* = 213 Age:18–42 yrs-old Molars with deep caries *n* = 525	Direct	A. Theracal B. Dycal C. Calcihyd D. Biodentine E. BioMTA+	**Clinical success** At 1 month: A = 100/105; B = 102/105; C = 98/105; D = 97/105; E = 98/105 At 6 months: A = 87/105; B = 81/105; C = 74/105; D = 88/105; E = 90/105 At 12 months: A = 77/105; B = 76/105; C = 68/105; D = 84/105; E = 90/105 At 3 years: A = 76/105; B = 73/105; C = 64/105; D = 83/105; E = 89/105	ND
**Rahman (2021)** **Randomized clinical trial**	Participants *n* = 55 Age:7–15 yrs-old Posterior teeth with deep caries *n* = 60 Dropouts:2 teeth per group *n* = 6 Participants at the end of the follow-up *n* = 54	Indirect	A. Theracal B. Dycal C. Biodentine	**Clinical success** At 3 weeks: A = 20/20; B = 19/19; C = 19/19 At 3 months: A = 19/19; B = 19/19; C = 19/19 At 6 months: A = 19/19; B = 18/19; C = 19/19 At 12 months: A = 19/19; B = 17/19; C = 18/19 At 18 months: A = 18/18; B = 15/18; C = 18/19 At 24 months: A = 18/18; B = 14/18; C = 17/18	Pain
**Covaci (2022)** **Observational study**	Participants *n* = 95 Age:34.66 ± 11.15 yrs-old Anterior teeth with deep caries *n* = 26 Posterior teeth with deep caries *n* = 69 Pulpar exposure *n* = 25 Non-pulpar exposure *n* = 70	Direct and indirect	A. TheraCal B. Calcimol C. Life Kerr	**Clinical success** At 6 months: A = 46/50; B = 26/31; C = 14/14	No
**Mahapatra (2022)**	Participants *n* = 21 Age:17–40 yrs-old Teeth with deep caries *n* = 28	Indirect	A: Theracal B. Prevest Cal	**Clinical success** At 6 months: A = 14/14; B = 13/14	Pain

ND: No data available. TheraCal (light-cured calcium silicate) Bisco Inc., Schaumburg, IL, USA. Dycal (self-cured calcium hydroxide) Dentsply-Sirona, Charlotte, NC, USA. Biner (light-cured hydroxy calcium phosphate). META-BIOMED, Chungcheongbuk-do, Korea. Calcihyd (light-cured calcium hydroxide) Dr. Roberts’, Istanbul, Turkey. Prevest Cal (light-cured calcium hydroxide) DenPro, Brussels, Belgium. MTA-Angelus, (mineral trioxide aggregate) Londrina, Paran, Brazil. Biodentine (self-cured calcium silicate) Septodont, Saint-Maur-des-Fosse’s France. ProRoot aggregate) Dentsply Tulsa, Johnson City, TN, USA. MTA PPH MTA (mineral trioxide aggregate) PPH CERKAMED, Stalowawola, Poland. BioMTA+ (modified tricalcium silicate) Cerkamed, Stalowa Wola, Poland. Calcimol LC (light-cured calcium hydroxide) Voco GmbH, Cuxhaven, Germany. Life Kerr AC (self-cured calcium hydroxide) Kerr, Orange, CA, USA.

**Table 5 biomimetics-07-00211-t005:** Quality assessment of the included studies in the systematic review.

Certainty Assessment							No of Patients		Effect		Certainty	Importance
**№ of Studies**	**Study Design**	**Risk of Bias**	Inconsistency	Indirectness	Imprecision	Other Considerations	Light-Cured CS	Comparator	Relative (95% CI)	Absolute (95% CI)		
Clinical success direct pulp capping, permanent teeth at 6 months												
3	Clinical trial	very serious	not serious	not serious	not serious	none	103/130 (79.2%)	106/130 (81.5%)	OR 0.86 (0.46 a 1.60)	24 fewer per 1000 (from 145 fewer to 61 more)	⨁⨁◯◯ Low	CRITICAL
Clinical success indirect pulp capping permanent teeth 6 months												
2	Clinical trial	serious	not serious	not serious	not serious	none	33/33 (100.0%)	31/33 (93.9%)	OR 3.19 (0.32 a 32.32)	41 higher per 1000 (from 107 fewer to 59 more)	⨁⨁⨁◯ Moderate	CRITICAL
Clinical success indirect pulp capping, permanent teeth 12 months												
2	Clinical trial	serious	not serious	not serious	not serious	none	128/130 (98.5%)	130/134 (97.0%)	OR 1.97 (0.35 a 10.94)	14 higher per 1000 (from 51 fewer to 27 more)	⨁⨁⨁◯ Moderate	CRITICAL

CI: Confidence interval; OR: Odds ratio.

## Data Availability

Data sharing not applicable–no new data generated.

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
