# Peer review of "Clinical Efficacy of Biomimetic Bioactive Biomaterials for Dental Pulp Capping: A Systematic Review and Meta-Analysis"

_biomimetics, 2022, doi:10.3390/biomimetics7040211_

Round 1

Reviewer 1 Report

This manuscript entitled “Clinical efficacy of biomimetic bioactive biomaterials for dental pulp capping: A systematic review and meta-analysis” (biomimetics-1981491) is a systematic review to evaluate Theracal-LC in comparison with conventional calcium silicate cement and calcium hydroxide in terms of clinical success.

General Concern

The authors tried to evaluate Theracal-LC comparing with calcium hydroxide or calcium silicate cement as a pulp capping material using meta-analysis. The results looked reasonable, but some points should be addressed. Such trials should be appreciated and the authors had better continue these works.

These are the specific concerns when reading the MS:

INTRODUCTION section:

- I felt the authors were not fair to pulp capping materials. It seemed that they might support Theracal. This can make the manuscript not so fair.

METHODS section:

- Indirect pulp capping were included in this study. But, when performing indirect pulp capping, few dentists may use pulp capping materials without specific purpose like step-wise excavation. I felt indirect pulp capping can be excluded. If the authors exclude indirect pulp capping, the number of the article can be too small to analyze. This can indicate this material may be too early to be meta-analyzed.

- Observation period should be longer.

Author Response

To whom it may concern -

Dear Colleague,

RE – Response to Reviewer 1 Comments

(All changes in the manuscript are highlighted with yellow colour)

We would like to thank the reviewer for spending time reviewing the article and providing constructive feedback. Having read the comments and consulted with the co-authors we have now made the appropriate amendments. We hope that the revised version meets your approval.

Comments

Responses

1- General Concern

The authors tried to evaluate Theracal-LC comparing with calcium hydroxide or calcium silicate cement as a pulp capping material using meta-analysis. The results looked reasonable, but some points should be addressed. Such trials should be appreciated and the authors had better continue these works.

These are the specific concerns when reading the MS:

INTRODUCTION section:

- I felt the authors were not fair to pulp capping materials. It seemed that they might support Theracal. This can make the manuscript not so fair.

The introduction has been modified according to this comment.

2- METHODS section:

- Indirect pulp capping were included in this study. But, when performing indirect pulp capping, few dentists may use pulp capping materials without specific purpose like step-wise excavation. I felt indirect pulp capping can be excluded. If the authors exclude indirect pulp capping, the number of the article can be too small to analyze. This can indicate this material may be too early to be meta-analyzed.

Thank you for pointing this out.

We totally agree with your comment; However, the paradigm is changing, as Bjørndal et al state: “recent consensus reports have stated that the complete or nonselective carious removal is now overtreatment”. Radiographs of deep carious lesions should be analyzed with the diagnostic data of the patient to determine if a tooth should be treated conservatively by avoiding pulp exposure from the start of treatment [1]. The current tendency attempt to preserve the pulpal integrity avoiding pulpal exposure [2]. As such, it is important to provide information about the available biomaterials for indirect pulp capping. Also, other systematic reviews have addressed indirect pulp capping. Besides, other authors of recent systematic reviews with meta-analysis have also performed forest plots of indirect pulp capping [3-5].

3- Observation period should be longer.

We completely agree with your concern that the studies should have longer follow-up time intervals.

In the Discussion section, we highlighted this observation:

.…many of the included studies had very short follow-up time intervals (<6 months). It has been reported that the follow-up time should be at least 6 months and that there is a slight difference in the outcome at 6 months and at 2 years, that is, the outcome at 6 months indicates the most likely outcome at 2 years [3].

Yours sincerely,

Authors

References

1.Bjørndal, L.; Simon, S.; Tomson, P.L.; Duncan, H.F. Management of deep caries and the exposed pulp. International Endodontic Journal 2019, 52, 949-973, doi:https://doi.org/10.1111/iej.13128.

2.Innes, N.P.T.; Frencken, J.E.; Bjørndal, L.; Maltz, M.; Manton, D.J.; Ricketts, D.; Van Landuyt, K.; Banerjee, A.; Campus, G.; Doméjean, S.; et al. Managing Carious Lesions: Consensus Recommendations on Terminology. Advances in Dental Research 2016, 28, 49-57, doi:10.1177/0022034516639276.

3.Stratigaki, E.; Tong, H.J.; Seremidi, K.; Kloukos, D.; Duggal, M.; Gizani, S. Contemporary management of deep caries in primary teeth: a systematic review and meta-analysis. European Archives of Paediatric Dentistry 2022, doi:10.1007/s40368-021-00666-7.

4.Santos, P.S.D.; Pedrotti, D.; Braga, M.M.; Rocha, R.O.; Lenzi, T.L. Materials used for indirect pulp treatment in primary teeth: a mixed treatment comparisons meta-analysis. Braz Oral Res 2017, 31, e101, doi:10.1590/1807-3107/2017.vol31.0101.

5.Tong, H.J.; Seremidi, K.; Stratigaki, E.; Kloukos, D.; Duggal, M.; Gizani, S. Deep dentine caries management of immature permanent posterior teeth with vital pulp: A systematic review and meta-analysis. Journal of Dentistry 2022, 124, 104214, doi:https://doi.org/10.1016/j.jdent.2022.104214.

Reviewer 2 Report

In this manuscript entitled “Clinical efficacy of Bioactive Biomaterials for Dental Pulp Capping: A Systematic Review and Meta-Analysis,” the authors revealed the efficacy of TheraCal LC and other calcium silicate-based biomaterials. Several in vitro studies with cell culture has showed that TheraCal LC has cytotoxicity due to the resin component. However, the authors show the material is not inferior to other materials in clinical trials.

I think this manuscript is informative.

Minor revision

Page 1, Line 44: “Instead” should be change to “instead.”

Page 2, Line 52: The authors should at least cite the references to clarify the “One study.” 

Page 5, Line 139: “Table 2” is not correct.

Page 6, Line 181-182: Please show the difference between pulp “destruction” and pulp “disorganization.”

Author Response

To whom it may concern -

Dear Colleague,

RE – Response to Reviewer 2 Comments

(All changes in the manuscript are highlighted with yellow colour)

We would like to thank the reviewer for spending time reviewing the article and providing constructive feedback. Having read the comments and consulted with the co-authors we have now made the appropriate amendments. We hope that the revised version meets your approval.

Comments

Responses

1- Page 1, Line 44: “Instead” should be change to “instead”.

Thank you for raising this point.

The word had been deleted since it was out of the context.

2- Page 2, Line 52: The authors should at least cite the references to clarify the “One study”.

The paragraph was modified to clarify the meaning.

However, there is no consensus about the ideal biomaterial for pulp capping. A multinational questionnaire-based survey performed by Stangvaltaite et al. [5] found that dentists who read scientific articles on the subject would consider, in the first instance, performing direct pulp capping instead of endodontic treatment (Odds Ratio=2.1, 95% confidence interval, 95% CI=1.3, 3.2). This study also revealed that about 50% of dentists use self-cured or light-cured calcium hydroxide for direct pulp capping. Ali et al. [6] found that the biomaterial preferred for pulp capping was TheraCal LC (39.7%), followed by calcium hydroxide (33.9%), MTA (21.55%), and Biodentine (5.0%); nevertheless, the use of calcium silicate-based biomaterials was significantly associated with endodontists respondents.

3- Page 5, Line 139: “Table 2” is not correct.

It has been changed.

4- Page 6, Line 181-182: Please show the difference between pulp “destruction” and pulp “disorganization.”

The content was changed according to this comment.

In terms of the histologic findings, none of the teeth treated with Biodentine showed disorganization of the entire pulp tissue, whereas 22.2% of TheraCal LC-treated and 44.4% of ProRoot MTA–treated cases showed disorganization of the entire pulp tissue.

However, there are no  major difference. The authors of the study classified the Pulp tissue organization and morphology in 3 categories: Normal or almost normal pulp tissue morphology, Disorganization of pulp tissue beneath the cavity, and Disorganization of entire pulp tissue.

Yours sincerely,

Authors

Reviewer 3 Report

This review and meta-analysis is about Theracal and its efficiancy for direct pulp capping. Introduction, justification, Search strategy, methods are well designed and described. As expectable, data suggest no difference but there is still need for more evidence.

English grammar needs checking by a native speaker.

In general, this is a well excecuted review research.

Author Response

To whom it may concern -

Dear Colleague,

RE – Response to Reviewer 3 Comments

(All changes in the manuscript are highlighted with cyan colour)

We would like to thank the reviewer for spending time reviewing the article and providing constructive feedback. Having read the comments and consulted with the co-authors we have now made the appropriate amendments. We hope that the revised version meets your approval.

Comments

Responses

1- English grammar needs checking by a native speaker.

Many thanks for this comment.

The manuscript has been reviewed and modified according to this comment.

Yours sincerely,

Authors

Round 2

Reviewer 1 Report

I have understood what the authors insisted.

I accepted the reason why they included indirect pulp capping in the manuscrpit.

However, response of the authors concerning the "unfair" Theracal recommendation was not enough for me to accept. I still feel unfair and the readers can be feel similar.

The authors' opinion for observation period "6 months outcomes are slightly different from 2 years outcome" from single systematic review may not be consensus among the researchers.

Author Response

We would like to thank the reviewer for taking the time and effort necessary to review the manuscript. We sincerely appreciate all valuable comments and suggestions, which helped us to improve the quality of the manuscript.

Reviewer: However, response of the authors concerning the "unfair" Theracal recommendation was not enough for me to accept. I still feel unfair and the readers can be feel similar.
Reply: Thank you for your important comments, and we understand the "unfair" statement. Therefore, the sentence "Ali et al. [6] found that the biomaterial preferred for pulp capping was TheraCal LC (39.7%), followed by calcium hydroxide (33.9%), MTA (21.55%), and Biodentine (5.0%); nevertheless, the use of calcium silicate-based biomaterials was significantly associated with endodontists respondents." has been changed to " Ali et al. [6] found that the biomaterial preferred was calcium silicate-based for direct and indirect pulp capping procedures." to address the reviewer's comment.

Reviewer: The authors' opinion for observation period "6 months outcomes are slightly different from 2 years outcome" from single systematic review may not be consensus among the researchers.
Reply: The sentence  "6 months outcomes are slightly different from 2 years outcome" has been removed from the revised manuscript to address the reviewer's comment.

Round 3

Reviewer 1 Report

I don't feel the response is enough.